# Bio-Oriented Synthesis and Molecular Docking Studies of 1,2,4-Triazole Based Derivatives as Potential Anti-Cancer Agents against HepG2 Cell Line

**DOI:** 10.3390/ph16020211

**Published:** 2023-01-30

**Authors:** Naheed Akhter, Sidra Batool, Samreen Gul Khan, Nasir Rasool, Fozia Anjum, Azhar Rasul, Şevki Adem, Sadaf Mahmood, Aziz ur Rehman, Mehr un Nisa, Zainib Razzaq, Jørn B. Christensen, Mohammed A. S. Abourehab, Syed Adnan Ali Shah, Syahrul Imran

**Affiliations:** 1Department of Biochemistry, Faculty of Life Science, Government College University Faisalabad, Faisalabad 38000, Pakistan; 2Department of Chemistry, Drug Design and Medicinal Chemistry Laboratory, Faculty of Physical Science, Government College University, Faisalabad 38000, Pakistan; 3Department of Zoology, Faculty of Life Sciences, Government College University Faisalabad, Faisalabad 38000, Pakistan; 4Department of Chemistry, Faculty of Sciences, Çankırı Karatekin University, 18100 Çankırı, Turkey; 5Department of Chemistry, Government College University, Lahore 54000, Pakistan; 6Department of Chemistry, University of Lahore, Lahore 40100, Pakistan; 7Department of Chemistry, Faculty of Science, University of Copenhagen, 2100 Copenhagen, Denmark; 8Department of Pharmaceutics College of Pharmacy, Umm Al-Qura University, Makkah 21955, Saudi Arabia; 9Faculty of Pharmacy, Universiti Teknologi MARA Cawangan Selangor Kampus Puncak Alam, Bandar Puncak Alam 42300, Selangor D. E., Malaysia; 10Atta-ur-Rahman Institute for Natural Product Discovery (AuRIns), Universiti Teknologi MARA Cawangan Selangor Kampus Puncak Alam, Bandar Puncak Alam 42300, Selangor D. E., Malaysia; 11Faculty of Applied Sciences, Universiti Teknologi MARA Shah Alam, Shah Alam 40450, Selangor D.E., Malaysia

**Keywords:** 2-(4-isobutylphenyl) propanoic acid, hepatocellular carcinoma, anti-cancer, 1,2,4-triazole, molecular docking, acetamides

## Abstract

Triazole-based acetamides serve as important scaffolds for various pharmacologically active drugs. In the present work, structural hybrids of 1,2,4-triazole and acetamides were furnished by chemically modifying 2-(4-isobutylphenyl) propanoic acid (**1**). Target compounds **7a–f** were produced in considerable yields (70–76%) by coupling the triazole of compound 1 with different electrophiles under different reaction conditions. These triazole-coupled acetamide derivatives were verified by physiochemical and spectroscopic (HRMS, FTIR, ^13^CNMR, and ^1^HNMR,) methods. The anti-liver carcinoma effects of all of the derivatives against a HepG2 cell line were investigated. Compound **7f**, with two methyl moieties at the ortho-position, exhibited the highest anti-proliferative activity among all of the compounds with an IC_50_ value of 16.782 µg/mL. **7f**, the most effective anti-cancer molecule, also had a very low toxicity of 1.190.02%. Molecular docking demonstrates that all of the compounds, especially **7f**, have exhibited excellent binding affinities of −176.749 kcal/mol and −170.066 kcal/mol to c-kit tyrosine kinase and protein kinase B, respectively. Compound **7f** is recognized as the most suitable drug pharmacophore for the treatment of hepatocellular carcinoma.

## 1. Introduction

In the 21st century, cancer and other infectious diseases are the most prevalent causes of death globally [1]. According to the World Health Organization (WHO), 11.5 million deaths are expected by 2030 due to cancer [2]. Among all types of cancer, hepatocellular carcinoma is among the leading causes of death, accounting for approximately 92% mortality rates worldwide [3,4]. Thus, the development of new anti-cancer drugs remains a huge clinical need for improving therapeutic efficacy and controlling cancer [5]. The development of multi-target anti-cancer agents is the major focus of researchers globally due to the different drawbacks associated with already-used chemotherapeutics such as undesirable side effects, a lack of selectivity, systemic toxicity, and the emergence of multidrug resistance [6,7,8,9]. Intensive efforts must be made to discover and develop new, effective, tailored –anti-cancer agents with better safety profiles and drug-like properties.

Structurally modified nitrogen-containing heterocyclic moieties have a broad spectrum of applications for the development of novel therapeutic drugs as shown in Figure 1 [10,11]. Approximately 75% of Food and Drug Administration-approved drugs are nitrogen-based moieties [12]. Nitrogen-containing heterocyclic compounds have been synthesised in large numbers in recent times. They exhibit anti-tubercular [13], anti-cancer [14], anti-fungal [15], anti-microbial [16], anti-viral [17], and other biological properties such as genotoxicity and lipid peroxidation [18], and anti-inflammatory properties [19].

Molecular hybridization is an easy and efficient method to combine various important drug pharmacophores. Our ongoing research focuses on the design and synthesis of pharmacologically active, diverse polyvalent scaffolds as anti-cancer agents. The versatile nature of 1,2,4-triazole has been reported to be of great importance in medicinal chemistry, such as for its anti-cancer [20], anti-fungal [21], anti-bacterial [22], anti-microbial, and anti-tumor properties [23], as well as pyrophosphatases and phosphodiesterase [24]. Acetamide has been identified as the most significant pharmacophore of anti-cancer drugs [25].

On this basis, we have created a hybrid of acetamide and 1,2,4-triazole pharmacophore by chemical derivatization of 2-(4-isobutylphenyl)propanoic acid in an attempt to avoid tumor progression. On each side of triazole and acetamide, we developed a molecular framework with hydrophobic aryl rings. It improves the solubility of the drug and its candidacy as a good drug molecule. Earlier, we reported the synthesis of various structural hybrids of oxadiazole-based acetamide [26,27,28,29], and it has been proven from the literature that heterocycle-based compounds possess good anti-cancer activity [30]. Thus, in an extension of our earlier research on heterocycles, –COOH group of 2-(4-isobutylphenyl)propanoic acid was cyclized into a 1,2,4-triazole ring 4.

## 2. Results

### 2.1. Chemistry

In the current study, 2-(4-isobutylphenyl)propanoic acid has been chemically modified with improved clinical utility. Different *N*-arylated 5-aryl-1,2,4-triazole-coupled acetamides (**7a–f**) have been synthesized in good yields by replacement of the H group of SH with various electrophiles.

Figure 1 depicts the synthetic route of final compounds **7a–f**. The Fischer esterification method was used to create compound 2 by refluxing compound 1 with absolute CH_3_OH at 76 °C for 3–4 h [31,32]. Compound 2 was slowly refluxed at 76 °C for 3–4 h with hydrazine hydrate [32] in CH_3_OH and yielded 2-(4-isobutylphenyl)propane hydrazide (3). Molecule **3** was converted into its respective 5-(1-(4-isobutylphenyl) ethyl)-1,2,4-triazole-2-thiol (4) by slowly heating it at 225 °C for 3–6 h with methyl isothiocyanate in 10% NaOH and absolute CH_3_OH. Upon completion, the reaction was acidified to pH 4–5 with conc. HCl. Upon acidification, precipitates appeared that had been separated as compound 4 by the process of filtration. At room temperature, compound 4 was treated with various *N*-arylated aralkyl/alkyl/aryl 2-bromoacetamides (**6a–f**) along with DCM and NaH as catalysts. The structures of triazole-coupled acetamide scaffolds were verified by physiochemical and spectroscopic (HRMS, FTIR, ^13^CNMR, and ^1^HNMR) methods.

In the ^1^HNMR spectrum, –NH protons of the acetamide were the most deshielded and their chemical shift value was observed around 10.27–9.64 ppm. In propanoic acid, the protons of the aliphatic region were among the most shielded, with values ranging from 0.86–0.84 ppm. The presence of acetamide was confirmed by the appearance of signals around 4.04 ppm for the CH_2_ group and in the range of 10.27–9.64 ppm for the NH group. In ^13^CNMR, the appearance of C=O signals in the range of 164.15–168 ppm confirmed the synthesis of acetamide. The presence of a triazole ring in the final derivatives was also confirmed by the appearance of signals in the range of 158.15 ppm and at 29.58 ppm for N-CH_3_. In the ^1^HNMR spectrum, signals for N-CH_3_ were observed around 3.28 ppm. A distinctive peak around 22.0 ppm corresponded to the protons of two -CH_3_ carbon nuclei. Signals for the CH_2_ groups were observed between 44.18 to 20.19 ppm. By introducing some electron-withdrawing substitutions and comparing them to electron-donating group substitutions, we reported structure–activity relationships for the phenyl group. 3,5-disubstituted triazole nuclei have a versatile nature and are important in the pharmaceutical industry. On the basis of their medicinal importance, the anti-cancer activities of all of the compounds were checked.

### 2.2. Anti-proliferative Potential

The anti-hepatocellular activity of afforded *N*-arylated 1,2,4-triazole coupled acetamides (**7a–f**) was evaluated via MTT assay and these structural hybrids were screened against a liver cancer HepG2 cell line [33]. All of the compounds demonstrated mild to outstanding anti-cancer activity, as shown in Table 1. Among all of the compounds, **7f**, with two methyl groups at positions 2 and 6 of the phenyl ring, displayed the best anti-cancer potential with IC_50_ = 16.782 µg/mL. Compound **7a**, which contains a methyl at position 2 of the phenyl ring, also displayed good anti-cancer activity with an IC_50_ value of 20.667 µg/mL but less than **7f**. Compounds **7b**, **7c**, and **7e** also exhibited a significant anti-cancer effect but less than **7f** and **7a.** Compound **7d**, with an electron-withdrawing Cl substituent, displayed the lowest anti-hepatocellular activity with a 39.667 µg/mL IC_50_ value.

The cell viability of all of the compounds was further evaluated using various concentrations (3.125–200 µg) to test the dose response and % inhibition relationship as shown in Table 2.

Figure 2 shows that compounds **7f** and **7a** produced the best results at a 25 µg/mL dose among all of the compounds.

The dose response and % inhibition of the most potent compound, **7f**, was checked at various concentrations (Figure 3).

### 2.3. Hemolytic Activity Potential

Hemolytic activity was investigated by the reported method [34] using Triton-X-100 as a standard. All of the triazole-based acetamide derivatives showed very low cytotoxicity, as shown in Table 1. Molecule **7f** also presented very low toxicity at 1.19 ± 0.02 relative to reference Triton-X-100. All of the other compounds had moderately good hemolytic activity. Compounds **7a** (2.46%), **7b** (4.43%), **7c** (4.32%), **7e** (4.19%), and **7d** (7.33%) also exhibited low cytotoxicity.

### 2.4. Structure–Activity Relationship of ***7a–f***

The anti-hepatocellular potential of all of the derivatives, **7a–f**, was evaluated against a HepG2 cancer cell line at various concentrations via MTT assay. The most potent derivative was **7f**, which contains two methyl groups at ortho-positions of the phenyl ring (IC_50_ value of 16.782 µM). Compound **7d**, which contained an electron-withdrawing Cl-group in an orthogonal position, showed the lowest anti-cancer potency with an IC_50_ value of 39.667 µM. The anti-proliferative activity of all of the derivatives was decreased in the following order: **7f** > **7a** > **7b** > **7c** > **7e** > **7d**. This proves that the attachment of an electron-donating CH_3_ group at the ortho-position increases the anti-cancer activity of the compounds. Based on the results of the SAR of *N*-arylated 5-aryl-1,2,4-triazole-coupled acetamide scaffolds 7a–f, it was determined that the -CH_3_ motif at the ortho-position of the phenyl ring improved the anti-proliferative potential of the compounds.

### 2.5. Molecular Docking

Molecular docking screenings were carried out to theoretically predict the most promising protein targets of compounds by molecular docking to some cancer targets. Five major targets in the treatment of cancer have been identified: human Aurora B kinase, phosphatidylinositol 3-kinase alpha (PI3Kalpha), the signal transducer and activator of transcription 3 (STAT3), protein kinase B (Akt), and c-kit tyrosine kinase (c-Kit). The web page http://www.swisstargetprediction.ch/ (Accessed on 7 November 2022) was used to study the potential anti-cancer effect of new synthesized molecules [35]. The results suggest that molecules may be effective against kinase targets.

Some important kinases and important targets in cancer treatment were determined by the literature review. c-kit tyrosine kinase endorses cellular proliferation by activating signal transduction mechanisms in response to stem cell factor adhesion [36]. Akt plays a vital role in internal cell signalling by accelerating cellular survival and proliferation. Its path becomes irregular during cancer [37]. Aurora kinase B regulates the cell cycle and is ubiquitously expressed in cancerous cells. [38]. The insulin-like growth factor-1 receptor (IGF-1R) has a vital role in cells in conjunction with PI3K–AKT and Ras–Raf–MEK signalling cascades, which control proliferation and apoptosis within cells. It is considered an important therapeutic target because of its deregulation of solid tumor types [39]. Phosphatidylinositol 3-kinase alpha is an intracellular lipid kinase that regulates cell survival, development, proliferation, and metabolism. It has been linked to a number of human cancers [40]. STAT3 is a secret transcription factor; it is regarded as an appealing target of anti-cancer therapeutics [41].

Table 3 shows the selected targets, grid box coordinates, their protein data bank codes, and Moldock scores. The docking findings demonstrate that molecules have the ability to affect a variety of targets. c-kit and Akt specifically are anticipated to have a high binding potential with cancer therapeutic targets, as demonstrated in Table 3. 

Figure 4 demonstrates 2D and 3D diagrammatic representations displaying the molecular bindings between the reference ligand SIT and the active pocket of c-Kit tyrosine kinase.

The docking findings demonstrate that all of the compounds have good binding affinity to kinase protein. The protein c-kit tyrosine kinase is expected to interact strongly with cancer-targeted therapeutics. Table 4 shows the docked complexes’ categories, modes of interactions, binding affinities, and by-products. Each ligand’s hydrophobic contacts and hydrogen bonding interactions were evaluated within the receptor protein’s binding pocket.

Compound **7f** bonded to c-kit tyrosine kinase with the most suitable binding pose and a low binding energy of −176.749 kcal/mol. It formed an H-bond with Asp810 and Glu640. Hydrophobic interactions occurred such as the pi-sigma bond with Thr670, His790, and Val643. Other hydrophobic interactions involved alkyl interactions with Val603, Val668, Leu644, Leu783, and Ile571, pi-sulfur interactions with Cys788 and Lys623, and amide-pi-stacked interactions with Cys809 residue. It also interacts with Val654, Ile789, Leu647, Ile808, Ile653, and Phe811 residues through van der Waals interaction. The 2D and 3D diagrammatic expressions of the binding interactions of **7f** and c-kit tyrosine kinase are shown in Figure 5.

Compound **7a** combines with c-kit tyrosine kinase with the most suitable binding poses with a low binding energy of −173.411 kcal/mol. Hydrophobic interactions of **7a** involved a pi-sigma bond with Leu595 and Leu644 and alkyl interactions with Cys673, Cys809, Ala621, Val654, Tyr672, and Phe811. It also interacts with Gly676, Leu799, Asp810, Leu647, Leu783, His790, Ile808, Glu640, Thr670, Val603, and Asp677 residues by van der Waals interactions. The 2D and 3D diagrammatic expressions of the binding interactions of **7a** and c-kit tyrosine kinase are shown in Figure 6.

The docked conformations of reference ligand X39 and protein kinase B were investigated in order to calculate qualitative estimates and the molecular basis of the analyzed bioactive molecules. Figure 7 shows 2D and 3D diagrammatic representations of the molecular bindings between the reference ligand X39 and kinase B protein’s active pocket.

According to the docking findings, all triazole-coupled acetamide scaffolds have a significant ability to influence the protein kinase B moiety. Akt is anticipated to have a high affinity for carcinoma therapeutic targets. Table 3 shows the docked complexes’ binding interactions, classifications, kinds of interactions, and interacting residues. Each ligand’s hydrophobic contacts and hydrogen bonding interactions were assessed within the receptor protein’s binding site. Table 5 describes the ligand conformations that demonstrated the greatest biological activity, as well as their suitable interactions in the receptors.

Compound **7a** had the most suitable binding poses with a binding energy of −166.843 kcal/mol to protein kinase B. Compound **7a** had the most suitable binding poses with a low energy of −166.843 kcal/mol to the catalytic site of protein kinase B. **7a** bonded to protein kinase B via H-bonding with Gly164, Phe163, and Thr162, a carbon–hydrogen bond with SerC9, pi-sigma interaction with Gly161, and pi–sulfer interaction with Asp293. It bonded with residues Leu183, val166, and Lys181 via hydrophobic alkyl interactions. Asn280, Lys277, Glu279, ArgC6, Lys160, Glu236, Gly159, Lys165, and ThrC8 residues involve van der Waals interactions among protein kinase B and compound **7a** (Figure 8).

**7f** and the protein kinase B complex is stabilized by H-bonding and van der Waals interactions. **7f** combines with the active site of the protein via H-bonding with Gly161, ArgC6, and Asp293. Hydrophobic bonds are involved with alkyl interactions with Lys181 and Leu183 and pi-sigma interactions with Val166 residues. Figure 9 demonstrates 2D and 3D diagrams displaying the most suitable binding interactions between **7f** and the active pocket of the kinase B protein.

## 3. Materials and Methods

### 3.1. General

In the present research, all of the starting materials were of analytical grade and were purchased from Alfa Aesar or Sigma Aldrich. 2-(4-isobutylphenyl) propanoic acid was used as a starting material. A Stuart SMP10 melting point apparatus was used for determining the melting point of all of the derivatives. The structures of all of the triazole-based scaffolds were confirmed by spectroscopy and physiochemical methods. An FT-IR spectrophotometer (4000–400 cm^−1^) by BRUKER was used at the Hi-Tech Lab, GC University, Faisalabad. ^1^HNMR spectra were recorded on an Bruker Advance 500 MHz spectrophotometer using DMSO-d6, on 5 mm diameter tubes at the University of Copenhagen, Denmark. An Bruker Advance NMR spectrophotometer was used to record ^13^C NMR spectra at 75 MHz by using DMSO-d6, on 5 mm diameter tubes. The reactions were supervised by thin layer chromatography.

### 3.2. General Procedure for the Synthesis of Synthesized N-Arylated 5-Aryl-1,2,4-Triazole-Coupled Acetamide Scaffolds ***7a–f***

#### 3.2.1. Synthesis of Methyl 2-(4-Isobutylphenyl)Propanoate (**2**)

Compound **2** was prepared by the reported method [42]. The compound (2) was obtained as a pale yellow, oily liquid. Yield: (90%); b.p. 263–265 °C; IR (KBr) cm^−1^: 1736.32, 1203.36, 1162.83; ^1^H NMR (400 MHz, CDCl_3_) δ 7.15 (d, 2H, *J* = 8.0 Hz), 7.05 (d, 2H, *J* = 8.0 Hz), 3.67 (q, 1H), 3.60 (s, 3H), 2.40 (d, 2H, *J* = 8.0 Hz), 1.82 (m,1H), 1.44 (d, 3H, *J* = 8.0 Hz), 0.85 (d, 6H, *J* = 8.0 Hz). ^13^C NMR (101 MHz, CDCl_3_) δ 175.22 (C=O), 140.49 (C-1), 137.63 (C-4), 129.04 (C-2 & C-6), 127.31 (C-3 & C-5), 52.09 (OCH_3_), 44.99 (CH_2_-9), 40.11 (CH-7), 30.25 (CH-10), 22.21 (CH_3_-11 & CH_3_-12), 18.50 (CH_3_-8). HRMS (ESI+): m/z calculated for [(C_14_H_20_O_2_)+H]^+^: 220.1463; found: 220.1460. Element analysis: C, 76.30; H, 9.16% (Appendix A).

#### 3.2.2. Synthesis of 2-(4-Isobutylphenyl)Propanehydrazide (**3**)

Compound **3** was prepared by the reported method [43]. 2-(4-isobutylphenyl) propane hydrazide was separated as a white, crystalline solid. Yield: (88%); m.p. 77–78 °C; IR (KBr) cm^−1^: 3272.76, 2963.13, 1640.12, 1604.83, 1466.29, 1366.62, 906.66, 686.83. ^1^H NMR (400 MHz, CDCl_3_) δ 9.50 (s, 1H), 7.13 (d, 2H, *J* = 8.0 Hz), 7.06 (d, 2H, *J* = 8.0 Hz), 3.48 (d, 2H), 3.46 (q, 1H), 2.40 (d, 2H, *J* = 8.0 Hz), 1.81 (m,1H), 1.48 (d, 3H, *J* = 8.0 Hz), 0.84 (d, 6H, *J* = 8.0 Hz). ^13^C NMR (101 MHz, CDCl_3_) δ 175.24 (C=O), 140.49 (C-1), 137.59 (C-4), 129.57 (C-2 & C-6), 127.30 (C-3 & C-5), 44.96 (CH_2_-9), 40.11 (CH-7), 30.25 (CH-10), 22.19 (CH_3_-11 & CH_3_-12), 18.23 (CH_3_-8). HRMS (ESI+): m/z calculated for [(C_13_H_20_N_2_O)+H]^+^. 220.1576; found: 220.1574. Element analysis: C, 70.85; H, 9.16; N, 12.72% (Appendix A).

#### 3.2.3. Synthesis of 5-(1-(4- Isobutylphenyl)Ethyl)-1,2,4-Triazole -2-Thiol (**4**)

In the current study, methyl isothiocyanate and 2-(4-isobutylphenyl) propane hydrazide (0.02 mol) were dissolved in 10% KOH soln. in an equimolar amount. For 10–11 h, the mixture was set on refluxing at 95 °C. Thin-layer chromatography was used for monitoring the reaction. Upon completion, cold water was added to afford the precipitates of product. Water was used to filter and wash the precipitates. The precipitates were further purified with an ethanolic recrystallization process. The 5-(1-(4-isobutylphenyl)ethyl)-1,2,4- triazole-2-thiol scaffold was crystalized as an off-white solid. ^1^HNMR (400 MHz, CDCl_3_) δ 11.77 (s, 1H, SH), 7.05 (d, 2H, *J* = 8.0 Hz), 6.99 (d, 2H, *J* = 8.0 Hz), 3.96 (q, 1H), 3.18 (s, 3H), 2.39 (d, 2H, *J* = 8.0 Hz), 1.80 (m,1H), 1.62 (d, 3H, *J* = 8.0 Hz), 0.83 (d, 6H, *J* = 8.0 Hz). ^13^C NMR (101 MHz, CDCl_3_) 155.16 (C-1 & C-4), 141.43 (C-1), 137.11 (C-4), 129.94 (C-2 & C-6), 126.87 (C-3 & C-5), 45.03 (CH_2_-9), 37.60 (CH-7), 30.67 (CH-10), 30.15 (CH_3_-N), 22.22 (CH_3_-11 & CH_3_-12), 20.20 (CH_3_-8). HRMS (ESI+): m/z calculated for [(C_15_H_21_N_3_S)+H]^+^: 275.1456; found: 275.1454. Element analysis: C, 65.41; H, 7.69; N, 15.27; S, 11.65% (Appendix A).

#### 3.2.4. Synthesis of N–Aryl/Alkyl 2-Bromoroacetamides **6a–f**

Compounds **6a–f** were synthesized using the reported method [27]. In an RBF, 12.0 moles N-substituted alkyl/aryl amines (**5a–f**) were dissolved in 10.0 mL of 5% Na_2_CO_3_ solution. Bromoacetyl bromide (12.0 mmoles) was gradually added to the reaction mixture described above. Upon reaction completion, n-hexane was added to afford arylated derivatives as precipitates which were further purified with an ethanolic recrystallization process or column chromatography technique using ethyl acetate–petroleum ether (1:9).

#### 3.2.5. Synthesis of N-Arylated 5-(1-(4-Isobutylphenyl)Ethyl)-1,2,4-Triazole-2-yl- 2-Sulfanyl Coupled Acetamide Derivatives **7a–f**

Various *N*-arylated 5-(1-(4-isobutylphenyl)ethyl)-1,2,4-triazole-2-yl-2-sulfanyl-coupled acetamide compounds were prepared in good yield by thoroughly mixing 4 (0.02 mol) with an equimolar amount of *N*-alkyl/aryl 2-bromoacetamides **6a–f** using DMF and NaH (0.01 mol). Thin-layer chromatography was used for monitoring the reaction. Upon reaction completion, n-hexane was added to afford arylated derivatives as precipitates which were further purified with an ethanolic recrystallization process or column chromatography technique using ethyl acetate–petroleum ether (1:9).

#### 3.2.6. *N*-(2-Methylphenyl)-2-((5-(1-(4-isobutylphenyl)ethyl)-4-methyl-4H-1,2,4-triazol-3-yl)thio)Acetamide (**7a**) 

White, amorphous solid. Yield 73%, m.p 122–124 °C. IR: ν (cm^−1^): 3270, 1696, 1524, 1488, 1306, 1082, 756. ^1^HNMR (500 MHz, DMSO) δ 9.64 (s, 1H), 7.40 (d, *J* = 5.0 Hz), 7.20 (d, *J* = 5.0 Hz), 7.15 (t, 1H), 7.09–7.06 (m, 5H), 4.33–4.29 (q, 1H), 4.04 (s, 2H), 3.28 (s, 3H), 2.40 (d, 2H, *J* = 10.0 Hz), 2.15 (s, 3H), 1.83–1.78 (m, 1H) 1.60 (d, *J* = 5.0 Hz), 0.85 (d, *J*= 10.0 Hz). ^13^C NMR (126 MHz, DMSO) δ 165.93, 158.08, 149.15, 139.59, 139.36, 135.90, 131.21, 130.27, 129.27, 126.81, 125.92, 125.18, 124.32, 44.15, 37.11, 35.47, 29.99, 29.53, 22.14, 20.93, 17.67 (Appendix A). HRMS (ESI+): m/z calculated for [(C_24_H_30_N_4_OS)+H]^+^: 423.2119; found: 423.2214 (Appendix A). Analysis calculated for C_24_H_30_N_4_OS, C, 68.21; H, 7.16; N, 13.26; S, 7.59%. 

#### 3.2.7. *N*-(4-Bromo-2-Mthylphenyl)-2-((5-(1-(4-Isobutylphenyl)Ethyl)-4-Methyl-4H-1,2,4-Triazol-3-yl)Thio)Acetamide (**7b**)

Off-white, amorphous solid. m.p 122–124 °C. Yield 71%. IR: ν (cm^−1^): 3370, 1670, 1528, 1470, 1306, 659.93. ^1^HNMR (500 MHz, DMSO) δ 9.68 (s, 1H), 7.43 (s, 1H), 7.40 (d, *J* = 5.0 Hz, 1H), 7.34 (dd, *J* = 10.0 Hz, 1H), 7.08–7.05 (m, 4H), 4.33–4.2 (q, 1H), 4.04 (s, 2H), 3.27 (s, 3H), 2.40 (d, 2H, *J* = 10.0 Hz), 2.15 (s, 3H), 1.80–1.75 (m, 1H), 1.60 (d, *J* = 10.0 Hz, 3H), 0.85 (d, *J* = 10.0 Hz, 6H). ^13^C NMR (126 MHz, DMSO) δ 166.17, 158.09, 149.05, 139.53, 139.30, 135.34, 133.77, 132.67, 129.25, 128.71, 126.78, 125.93, 44.15, 37.10, 35.47, 30.00, 29.53, 22.14, 20.92, 17.40 (Appendix A). HRMS (ESI+): m/z calculated for [(C_24_H_29_BrN_4_OS)+H]^+^: 501.1324; found: 503.1313 (Appendix A). Analysis calculated for C_24_H_29_BrN_4_OS. Elemental Analysis: C, 57.48; H, 5.83; N, 11.17; S, 6.39. 

#### 3.2.8. *N*-(4-Ethylphenyl)-2-((5-(1-(4-Isobutylphenyl)Ethyl)-4-Methyl-4H-1,2,4-Triazol-3-yl)Thio)Acetamide (**7c**) 

Off-white, amorphous solid. Yield 75%, m.p 100–102 °C. IR: ν (cm^−1^): 3235, 1682, 1517, 1468, 1320, 695. ^1^HNMR (500 MHz, DMSO) δ 10.18 (s, 1H), 7.43–7.42 (d, 2H, *J* = 5.0 Hz), 7.14–7.13 (d, 2H, *J* = 5.0 Hz), 7.05–7.01, (m, 4H,) 4.32–4.29 (q, 1H), 3.98 (s, 2H), 3.26 (s, 3H), 2.39(d, 2H, *J* = 5.0 Hz), 1.81–1.76 (m, 1H), 1.59 (d, J = 12, 3H), 1.17–1.14 (t, 3H) 0.84 (d, *J* = 5.0 Hz, 6H). ^13^C NMR (126 MHz, DMSO) δ 165.44, 157.97, 149.05, 139.46, 139.35, 138.87, 136.40, 129.25, 127.91, 126.68, 119.14, 44.15, 37.84, 35.49, 29.97, 29.52, 27.54, 22.14, 20.89, 15.60 (Appendix A). HRMS (ESI+): m/z calculated for [(C_25_H_32_N_4_OS)+H]^+^: 437.2375; found: 437.2366 (Appendix A). Analysis calculated for C_25_H_32_N_4_OS, C, 68.77; H, 7.39; N, 12.83; S, 7.34.

#### 3.2.9. *N*-(2-Chlorophenyl)-2-((5-(1-(4-Isobutylphenyl)Ethyl)-4-Methyl-4H-1,2,4-Triazol-3-yl)Thio)Acetamide (**7d**)

White, amorphous solid. Yield 70%, m.p 123–125 °C. IR: ν (cm^−1^): 3330, 1691, 1515, 1452, 1315, 1081, 757. ^1^HNMR (500 MHz, DMSO) δ 9.90 (s, 1H), 7.75 (d, 1H, *J* = 10.0 Hz), 7.50 (d, 1H, *J* =10.0 Hz), 7.33–7.31 (t, 1H, *J* = 5.0 & 10.0 Hz), 7.21–7.18 (t, 1H, *J* = 5.0 & 10.0 Hz), 7.10–7.06 (m, 4H), 4.33–4.29 (q, 1H), 4.12 (s, 2H), 3.28 (s, 3H), 2.40(d, 2H, *J* = 5.0 Hz), 1.79–1.76 (m, 1H) 1.60 (d, *J* = 5.0 Hz, 3H), 0.84 (d, *J* = 10.0 Hz, 6H). ^13^C NMR (126 MHz, DMSO) δ 166.56, 158.12, 148.93, 139.65, 139.28, 134.53, 129.50, 129.27, 127.43, 126.81, 126.14, 125.14, 44.07, 37.07, 35.48, 30.04, 29.53, 22.13, 20.97 (Appendix A). HRMS (ESI+): m/z calculated for [(C_23_H_27_ClN_4_OS)+H]^+^: 444.1672; found: 444.1697 (Appendix A). Analysis calculated for C_23_H_27_ClN_4_OS, C, 62.36; H, 6.14; N, 12.65; S, 7.24.

#### 3.2.10. *N*-(Phenyl)-2-((5-(1-(4-Isobutylphenyl)Ethyl)-4-Methyl-4H-1,2,4-Triazol-3-yl)Thio)Acetamide (**7e**)

White, amorphous solid. Yield 74%, m.p 88–90 °C. IR: ν (cm^−1^): 3250, 1686, 1525, 1436, 1303, 1082, 757. ^1^HNMR (500 MHz, DMSO) δ 10.27 (s, 1H), 7.54 (d, 2H, *J* = 10.0 Hz), 7.32–7.29 (t, 2H, *J* = 5.0 & 10.0 Hz), 7.10–7.02 (m, 5H), 4.31–4.29 (q, 1H), 4.01 (s, 2H), 3.29 (s, 3H), 2.39(d, 2H, *J* = 5.0 Hz), 1.79–1.76 (m, 1H), 1.59 (d, *J* = 5.0 Hz, 3H), 0.85 (d, *J* = 10.0 Hz, 6H). ^13^CNMR (126 MHz, DMSO), δ 165.75, 158.10, 149.03, 139.60, 139.33, 138.68, 129.25, 128.74, 126.77, 123.42, 119.04, 44.10, 37.79, 35.41, 29.88, 29.49, 22.14, 20.89 (Appendix A). HRMS (ESI+): m/z calculated for [(C_23_H_28_N_4_OS)+H]^+^: 409.2044; found: 409.2054 (Appendix A). Analysis calculated for C_23_H_28_N_4_OS, C, 67.62; H, 6.91; N, 13.71; S, 7.85. 

#### 3.2.11. *N*-(2,6-Dimethylphenyl)-2-((5-(1-(4-Isobutylphenyl)Ethyl)-4-Methyl-4H-1,2,4-Triazol-3-yl)Thio)Acetamide (**7f**)

Off-white, amorphous solid. Yield 76%, m.p 150–152 °C. IR: ν (cm^−1^): 3272, 1640, 1516, 1445, 1388, 1081, 694. ^1^HNMR (500 MHz, DMSO) δ 9.68 (s, 1H), 7.12–7.02 (m, 7H), 4.40–4.23 (q, 1H), 4.06 (s, 2H), 3.29 (s, 3H), 2.29 (d, 2H, *J* = 10.0 Hz), 2.13 (s, 6H), 1.81–1.78 (m, 1H), 1.61 (d, *J* = 5.0 Hz, 3H), 0.85 (d, *J*= 10.0 Hz, 6H). ^13^C NMR (126 MHz, DMSO) δ 165.46, 157.98, 149.10, 139.60, 139.38, 135.06, 134.57, 129.26, 127.57, 126.83, 126.49, 44.15, 36.59, 35.46, 29.97, 29.54 (Appendix A). HRMS (ESI+): m/z calculated for [(C_25_H_32_N_4_OS)+H]^+^: 437.2375; found: 437.2366 (Appendix A). Analysis calculated for C_25_H_32_BN_4_OS, C, 68.77; H, 7.39; N, 12.83; S, 7.34.

### 3.3. Experimental Procedures for Biological Activities

#### 3.3.1. Cell Culture and Treatment

Human HepG2 liver cancer cell lines were cultured by Dulbecco’s modified Eagle’s medium. It is composed of 100 μg/mL streptomycin, 100 units/mL penicillin, and 10% FBS. A humidified atmosphere was provided for incubation at 37 °C with 5% CO_2_. The anti-hepatocellular therapeutic potential of triazole-based scaffolds was evaluated by dissolving its different concentrations in 0.05% DMSO.

#### 3.3.2. Evaluation of Cell Viability

An MTT assay was applied for evaluation of cell viability against the HepG2 cell line [44]. In short, different concentrations of new triazole-based scaffolds were incubated with HepG2 cell lines for 48 h. After incubation, 5 mg/mL of 10 µL MTT solution was added in each plate and they were further incubated at 37 °C for 4 h. The percentage of cell viability was calculated at 490 nm after the addition of 150 μL DMSO into a microplate reader (Thermo Scientific, Waltham, MA, USA).

#### 3.3.3. Hemolytic Activity Potential

Hemolytic activity was investigated by the reported method [45,46] using Triton-X-100 as standard.

### 3.4. Molecular Docking of Triazole-Coupled Acetamides

Docking experiments for all of the scaffolds were carried out in order to comprehend the potential interaction process of the synthesized anti-cancer compounds on the HepG2 cancer cell line. The website https://www.rcsb.org was used for drawing the structures of PI3Kalpha, Akt, c-kit tyrosine kinase, human Aurora B kinase, and STAT3 from the RCSB Protein Data Bank under the PDB IDs of 4FA6, 2X39, 1T46, 4AF3, and 6NJS, respectively [36,37,38,39,47]. ChemDraw 20.1.1 was used to create and reduce the 3D SDF structures of all of the compounds, which were then transferred to MarvinSketch. Prior to docking, the target proteins’ frameworks were evaluated, and errors in amino acid structures were rectified using Molegro Virtual Docker software [48]. The grid boxs’ centers were chosen to be the co-crystallized ligands of proteins. They re-docked in order to validate the in silico process. Molegro Virtual Docker was applied to dock active chemicals 10 times to the target proteins’ receptors. The sequences with the lowest interaction affinity and excellent connections with the targets were separated for further detailed analysis. The molecular bindings between the target and new derivatives were visualized in 2D using Discovery Studio Visualizer Software 2021.

## 4. Conclusions

A series of new anti-cancer compounds (**7a–f**) were synthesized in moderate to good yield (73–76%) by combining compound 4 with various electrophiles under different reaction conditions (Table 1, Figure 1). Because of its low bioavailability of 38–49%, Sorafenib necessitates a significant daily dose in cancer therapy. Sorafenib is a very costly medicine with many side effects. We have incorporated various electron-donating and electron-withdrawing groups into electrophiles to test structure–activity relationships at various concentrations. All of the molecules demonstrated medium to outstanding anti-cancer activity, comparable to sorafenib, which diversified according to aryl ring substitution, as shown in Table 1. These triazole-based acetamide derivatives also exhibited low cytotoxicity, with values ranging from 7.33% to 1.19% in comparison to the 100% cytotoxicity exhibited by the reference standard Triton X100. Compounds **7f** and **7a** showed the highest anti-cancer potential, with IC_50_ values of 16.782 µg/mL and 20.667 µg/mL, respectively. On the other hand, the triazole derivative containing an electron-withdrawing chloro moiety demonstrated the least anti-proliferative activity with an IC_50_ value of 39.667 µg/mL. The sequence of anti-cancer potential was found to be **7f** > **7a** > **7b** > **7c** > **7e** > **7d**. The anti-cancer potential of all of the compounds was further investigated by molecular docking studies and the results were in accordance with in-vitro studies. In silico studies have shown that the molecules have strong affinity for kinase targets. Molecules **7f** and **7a** have shown their anti-cancer effects, especially by affecting Akt and c-lit molecular targets. According to in silico modelling studies, **7f** has an outstanding docking score with the lowest binding energy of −170.066 kcal/mol, which is lower than the reference ligand X39 for protein kinase B (−130.624 kcal/mol). We concluded that compound **7f** contained electron-donating methyl groups at the 2 and 6 position of the aryl ring and showed good anti-cancer activity, low cytotoxicity, and good thrombolytic activity. Thus, compound **7f** might be utilized to synthesize new anti-cancer drugs in the near future.

## Data Availability

Data is contained within the article and Appendix A.

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
