# Peer review of "Bio-Oriented Synthesis and Molecular Docking Studies of 1,2,4-Triazole Based Derivatives as Potential Anti-Cancer Agents against HepG2 Cell Line"

_pharmaceuticals, 2023, doi:10.3390/ph16020211_

Round 1

Reviewer 1 Report

Manuscript Title: Bio-Oriented synthesis and molecular docking studies of 1,2,4-Triazole-based derivatives as potential Anticancer agents against HepG2 Cell Line.

Manuscript ID: Pharmaceutical-2077496

This manuscript entitled “Bio-Oriented synthesis and molecular docking studies of 1,2,4-Triazole based derivatives as potential Anticancer agents against HepG2 Cell Line” is mainly focused on the in-silico studies of anticancer drug 1,2,4-Triazole based derivatives. In the current situation in front of the world, there is a big challenge of cancer and its treatment. Hence present study could be best fitted in ongoing research on cancer. The present form of the manuscript is having potential data but there are a few mistakes that need to be corrected before moving further with the decision.

Comment 1: In scheme 1 it is better to label a, b, c, and the other images to make clear ideas about the whole scheme to new readers.

Comment 2: It is also beneficial to add the reaction conditions on the arrow in the scheme at the appropriate position. The addition of reaction conditions and proper labels could be very useful to understand the whole scheme of compound synthesis with ease.

Comment 3: On Page No. 4 Line No. 133 “liver cancer HepG2” could be replaced with “liver cancer HepG2 cell lines” for making a complete sentence. Proofread the whole manuscript and remove the errors and add necessary full forms as well as missing words to complete all the sentences with proper meaning.  

Comment 4: On page No. 5 Line No. 146 and 147 the word “%age” should be replaced by “%” the “age” seems to be meaningless. Also, make the same correction throughout the manuscript such as in Figure 2 also.

Comment 5: In the Figure 3 legend “Dose-response and % age inhibition relationship of all compound 7f at different concentration.” Should be replaced with “Dose-response and % inhibition relationship of compound 7f at different concentrations”

Comment 6: In Table 1 hemolytic activity of “Sorafenib” is not mentioned if possible it could be added for better comparison and making a complete Table.

Comment 7: In Figure 2 Dose-response and % inhibition relationship has been shown for all the compounds. The error bars showed are showing too many fluctuations are they right? Like it shows fluctuations like 80-100 for a compound as well as for compound 7f which shows negative % inhibition. How it is possible that there is negative % inhibition? Correct the graph for better understanding. However, the same is applied to Figure 3 error bars.

Comment 8: In the molecular docking image the reference ligand IT-46 should be labeled in the figure. The active site residues of C-Kit Tyrosine Kinase are properly labeled but the ligand is not labeled.

Comment 9: In the molecular docking image of Figure 5 The residue labels are not visible because it is too small. The image size or label size should be enlarged to see the labels clearly in the manuscript for a better understanding of docking interactions.

Comment 10: In Table 5 the label of the last column “Interactions Residues” could be changed to “interacting residues”

Comment 11: The important amino acids of Protein Kinase B (Akt) and c-Kit Tyrosine Kinase (c-Kit) involved in the bonding interactions with Compound 7f could be mentioned in the conclusion also for better projection of molecular docking studies.

Comment 12: Proofread the whole manuscript for grammar and punctuational mistakes.

However, the manuscript is having valuable data but still needs to be represented well for a better understanding of the reader. The manuscript needs major revision before considering further.

Reviewer 2 Report

In the submitted manuscript, the authors describe the design and synthesis of a hybrid of triazole and acetamide pharmacophores and investigates their potential as  anti-liver carcinoma effects against the HepG2 cell line. Coupling of various electrophiles with triazoles analogues afforded in the desired compounds with decent yield and the molecules were characterized using the various analytical and physiochemical methods. Their Hemolytic activity Potential, anti-proliferative potential, and IC50 of the compounds were also investigated along with molecular docking to see the potential of synthesized compounds for exploring the suitable protein targets in cancer treatment.

The introduction, results and discussion section are properly written, and authors have provided the experimental data to support their claims. However, the authors should address the following points before the manuscript can be considered publishable in Pharmaceuticals-

·      Reference 3  is from 2016, if there have been recent reports related to that, please cite them, so it is clear that what the authors wants to convey is still a critical issue with high mortality.

·      Figure 1 and Figure 3 are there in the manuscript but has never been referred in the text or referenced in discussion. Please explain?

·      Line 85: “and it is proven from the literature that these heterocycle-based 85 compounds possess good anticancer activity” add suitable reference to this claim.

·      Section 2.2. Antiproliferative Potential: Would be better for the reader if the authors can compare the anticancer activity with one or two of the reported molecules demonstrating excellent activity or even better if the authors compare it with the agents currently approved for used in clinic. Like the entry in table 1: Sorafenib.  Discuss the results with respect to sorafenib or another approved clinical anticancer agent.

·      Add the reaction temperature for each step in the scheme 1.

·      The reported yield for compound 7a-7f, is it the yield after recrystallization? Additionally, authors should also mention the overall yield range for compounds 7a-7f.

·      NMR and Mass spectroscopy characterization data of precursor compounds 6a-6f is missing. Please provide the full characterization data along with reaction yield for all of these compounds.

·      The integration in the NMR spectrum in Figure S4 for aromatic region is not there, please correct the NMR spectra and add a spectrum with all integration.

·      Most of the 13C NMR spectrum of compounds 7a-7f needs to not phased appropriately, please replace the spectra with the properly phased 13C NMR spectra.

Round 2

Reviewer 2 Report

Dear Authors,

   Thank you for providing the response to my comments and clarifying or adding the necessary text. I am satisfied with the the response and thank the authors for addressing each of the point raised.

I have one minor additional suggestions regarding the 13C NMR spectra of the compounds 7a-7f. I would suggest that the authors also add the expanded 13C NMR (110-180 ppm region), similar to what has been done for 1H NMR spectra.  The signals are not clearly visible due to the dominating signal from CDCl3. Please increase the peak height in the expanded spectra.

Thank You
